# High prevalence and co-infection of high-risk *Human Papillomavirus* genotypes among unvaccinated young women from Paraguay

**María Liz Bobadilla**[1,2]*, **Verónica Villagra**[1], **Violeta Ortiz**[1], **Gerardo Deluca**[3], **Vanessa Salete de Paula**[3]

**1** Laboratory of Immunology, Central Laboratory of Public Health/Minister of Public Health and Social Welfare, Asunción, Paraguay, **2** Laboratory of Molecular Virology and Parasitology, Oswaldo Cruz Institute, Oswaldo Cruz Foundation, Rio de Janeiro, Brazil, **3** Molecular Applications Laboratory, Faculty of Medicine, Northeast National University, Corrientes, Argentina

\* bobadillaml@gmail.com

## Abstract

Paraguay launched a human papillomavirus (HPV) vaccination program in 2013, so virological surveillance is important for measuring the impact of HPV vaccines. This study aimed to estimate the type-specific HPV frequency in unvaccinated sexually active women aged 18–25 years in the metropolitan area of Asuncion as a baseline for monitoring the HPV vaccination program. This study included 208 women, attending the Central Laboratory of Public Health between May 2020 and December 2021, were invited for testing through social networks and flyers at local health centers and higher education institutes. All participants who agreed to contribute to the study signed a free, prior, and informed consent form and answered a questionnaire that included basic demographic data and determining factors of HPV infection. Human papillomavirus detection and genotyping were conducted using the CLART HPV2 test (Genomica, Madrid, Spain) that allows the individual identification of 35 genotypes. 54.8% women were positive for any HPV type, with 42.3% positive for high-risk HPV (HR-HPV) types. Several factors were associated with HPV detection including the number of sexual partners, new sexual partners, non-use of condoms, and history of other sexual infections. Moreover, multiple infections were identified in 43.0% of the young women. We detected 29 different viral types present in both single and multiple infections. HPV-58 was the most commonly detected HPV type (14.9%), followed by HPV-16, HPV-51, and HPV-66 (12.3%). We estimated the prevalence of bivalent (16/18), quadrivalent (6/11/16/18), and nonavalent (6/11/16/18/31/33/45/52/58) vaccine types to be 8.2%, 13%, and 38%, respectively. These results reinforce the importance of surveillance studies and provide the first data regarding circulating HPV genotypes in the unvaccinated population in Paraguay, thus generating a baseline to compare future changes in the overall and type-specific HPV prevalence after HPV vaccination.

**Data Availability Statement:** There are ethical or legal restrictions on sharing data set, as data contain potentially sensitive information. The

contact information of Ethics Committee of Central Laboratory of Public Health of Minister of Public Health and Social Welfare is biblioteca. lcsp@mspbs.gov.py and cei.lcsp.py@gmail.com.

**Funding:** This study was supported by the National Council of Science and Technology (CONACYT) through the PROCIENCIA Program with resources from the Fund for Excellence in Education and Research (FEEI) (RESERCH 15-INV-200), Research, Education and Biotechnologies Applied to Health (IEBAS) Project – MERCOSUR Structural Convergence Fund (FOCEM), COF 03-11, Coordination for the Improvement of Higher Education Personnel (Coordenação de Aperfeiçoamento de Pessoal de Nível Superior - CAPES) - Finance Code 001, Fundação Carlos Chagas Filho de Amparo à Pesquisa do Estado do Rio de Janeiro (FAPERJ), Oswaldo Cruz Institute and Conselho Nacional de Desenvolvimento Científico e Tecnológico (CNPq).This study was also partially supported by the Coordenação de Aperfeiçoamento de Pessoal de Nível Superior – CAPES - Finance Code 001.

**Competing interests:** The authors have declared that no competing interests exist.

## Introduction

Human papilloma virus (HPV) is the most common sexually transmitted infection (STI) among sexually active young women, being acquired shortly after the first sexual intercourse, with the highest prevalence seen in women aged 25 years or younger [1, 2].

Based on epidemiological and biological data HPV 16, 18, 31, 33, 35, 39, 45, 51, 52, 56, 58, 59, 68, 73, and 82 should be considered carcinogenic, or high-risk types (IARC Group 1); HPV26, 53, 66, 67, 68, 70, 73, and 82 probably/possibly carcinogenic or probable/possible high-risk types (IARC Groups 2A and 2B) and HPV 6 and 11 non-carcinogenic, or low-risk types (IARC Group 3) [3].

Persistent infection with HPV, especially high-risk types, leads to the development of cervical cancer [4]. HPV-16 and HPV-18 are the most common carcinogenic types, responsible for almost 70% of all cervical cancers. Five additional high-risk types, 31, 33, 45, 52, and 58, are responsible for another 15% of all cervical and 11% of all HPV-associated cancers. HPV-6 and HPV-11 are the two primary "low-risk" types that cause anogenital warts [5].

Cervical cancer is the fourth most common cancer among women, with an estimated 604,000 new cases and 342,000 deaths worldwide in 2020, with nearly 90% of these occurring in low- and middle-income countries [6]. In Latin America and the Caribbean, approximately 60,000 new cases of cervical cancer and more than 31,000 deaths due to the disease occurred in 2020. In Paraguay, the incidence is much higher than that observed in other countries in the region, with incidence and mortality rates of 34.1 and 19.1 per 100,000 women, respectively [7].

Cervical cancer is a disease which reflects inequities among different populations depending on the availability of a national vaccination program and population-based cervical cancer screening, and access to quality treatment. In 2020, the World Health Organization (WHO) called for action to eliminate cervical cancer as a public health problem. With three key strategies and clear 2030 targets: 90% of girls fully vaccinated with HPV vaccine by 15 years of age, 70% of women screened using a high-performance test by 35 years of age and again by 45 years, and 90% of women identified with cervical disease treated [8].

The primary prevention approach focuses on preventing disease before it develops. WHO recommends including the HPV vaccine in the national immunization schedule. Currently, three HPV vaccines have been internationally approved: a bivalent (Cervarix), a quadrivalent (Gardasil), and a nonavalent (Gardasil-9) vaccine, consisting of type-specific HPV L1 virus-like particles (VLPs) that induce type-restricted protection. All three vaccines prevented HPV-16 and HPV-18 infection, with the quadrivalent Gardasil also protecting against HPV-6 and HPV-11, and the nonavalent Gardasil-9 targeting an additional five HPV types (HPV-31, 33, 45, 52, and 58) [9]. These vaccines may also have some cross-protection against other less common oncogenic types, such as HPV-31, 33, and 45 [10].

Secondary prevention, based on cervical screening program, is widely implemented. The adoption of Pap smear and HPV testing resulted in a significant increase in the diagnosis of cervical dysplasia and a significant decrease in cervical cancer, mainly in more developed countries. New screening strategies, such as HPV self-sampling, and the use of artificial intelligence in colposcopy assessment, must be disseminated in a near future [11].

Some markers, including p16 ink4a, p16, E-cadherin, Ki67, pRb, and p53, have been found to be useful in identifying intra-epithelial lesions that are more likely to develop into invasive lesions. Others, such as CEA, SCC-Ag, CD44, were developed to detect invasive forms. These biomarkers represent a promising diagnostic, prognostic and dynamic tool and can also be easily performed in the management of cervical cancer [12].

And tertiary prevention in cervical cancer includes its treatment with surgery (radical hysterectomy), radiotherapy, and systemic treatments (chemotherapy, bevacizumab, and

immunotherapy) [11]. Minimally invasive surgery has been studied and associated with advantages in the treatment of various gynecological cancers [13].

In Paraguay, the quadrivalent HPV vaccine was introduced into the national immunization program in 2013, prioritizing girls ages 9–10 years old with 0-1-6-month immunization schedule using school-based vaccination [14].

The detection of HPV infection can precede advanced stages of the disease, such as cervical intraepithelial neoplasia grade 3 (CIN 3), for a period of five to ten years; therefore, identifying prevalent HPV genotypes can provide earlier clues about the impact of the vaccine program [9, 15].

In Paraguay, primary screening aims to detect precursor lesions of cervical cancer in women over 30 years of age [14], making screening data inadequate for assessing the early impact of the vaccine. Consequently, establishing a national HPV surveillance program that includes a series of cross-sectional studies to assess the prevalence of HPV in cervical samples from young women would aid in identifying the impact of the vaccination program [9].

This study aimed to estimate the type-specific HPV frequency in unvaccinated women aged 18–25 years in the metropolitan area of Asuncion as a baseline for monitoring the HPV vaccination program. Here, we provide the first data about circulating genotypes in the unvaccinated population, generating a baseline to compare future changes in overall and type-specific HPV prevalence after HPV vaccination.

## Materials and methods

### Study population

This cross-sectional study was conducted between May 2020 and December 2021, using convenience sampling strategy, due to the national primary screening is mainly aimed at women aged 30 years and above [14]. Sexually active women aged 18–25 years, who had not been vaccinated against HPV were invited for screening through social networks and flyers at local health centers and higher education institutes. Exclusion criteria were pregnancy, nonsexual onset at the time of the study and have been received HPV vaccine. Women who voluntary agreed to participate signed a free, prior, and informed consent form and answered a questionnaire that included basic demographic data and determining factors of HPV infection (age, origin, occupation, educational degree, age at first sexual intercourse, number of sexual partners in the last 12 months, contraceptive methods, sexual habits, alcohol consumption, and tobacco use). 208 young women were included, and their samples were processed at the Central Laboratory of Public Health (LCSP), the National Reference HPV Laboratory in Paraguay.

### Clinical specimen collection

After a personal interview, specimens for HPV detection were collected from the endocervix/ectocervix using a cervical cytology brush (PAPETTEⓇ Cervical Cell Collector, WALLACH Surgical Devices, Trumbull, CT, USA). The samples were resuspended by shaking the brush in vials with 2.0 ml of sterile phosphate-buffered saline (PBS) to detach the collected cells. The brush was discarded, and the samples were kept at 8˚C until processing within 72 h. All samples were assigned an anonymous and unique patient code.

### DNA extraction from cervical samples

Aliquots of 1.0 ml were centrifuged at 12,000 rpm for 10 min to pellet the exfoliated cervical cells. After removing the supernatant, 1.0 ml of sterile water was added to resuspend and wash the cells. The samples were centrifuged again at 12,000 rpm for 10 min. The supernatants were

discarded, and the pellets were resuspended in a mix of 180 µl PBS and 20 µl proteinase K (20 µg/ml). The samples were vortexed and incubated for 1 h at 65˚C. Finally, HPV DNA was extracted using a CLART HPV 2 DNA Extraction/Purification Kit (Genomica, Madrid, Spain) according to the manufacturer's instructions, and 100 µl of eluted DNA was stored at -20˚C until use.

## HPV genotyping

Human papillomavirus detection and genotyping were conducted using the CLART HPV2 test (Genomica, Madrid, Spain) according to the manufacturer's instructions. Briefly, a 450 bp fragment in the HPV L1 region was PCR amplified using biotinylated primers. DNA extraction adequacy and PCR efficiency were tested by co-amplification of an 892 bp region of the cystic fibrosis transmembrane conductance regulator (CFTR) gene and a 1,202 bp fragment of a transformed plasmid. Amplicons were detected by hybridization in a low-density microarray with specific DNA probes to 35 HPV types (HPV 6, 11, 16, 18, 26, 31, 33, 35, 39, 40, 42, 43, 44, 45, 51, 52, 53, 54, 56, 58, 59, 61, 62, 66, 68, 70, 71, 72, 73, 81, 82, 83, 84, 85, and 89). The results were obtained using an automatic reader.

We used 5 µl of purified DNA for PCR amplification and the PCR products were labeled with biotin. The PCR products were denatured at 95˚C for 10 min and visualized using 5 µl of the denatured PCR products. Hybridization between amplicons and their specific probes was performed using probes immobilized in the wells of the hybridization plate. A streptavidin conjugate was added to the wells, which bound to the biotin-labeled PCR products, resulting in the formation of an insoluble peroxidase precipitate. The precipitate was analyzed using a Clinical Array Reader (Genomica, Madrid, Spain). Samples with invalid outcomes were retested, and the second result was considered conclusive.

## Statistical analyses

A descriptive analysis of the epidemiological characteristics of study participants was performed. All variables were recorded and coded in Microsoft excel 2013 and data were processed using the Epi Info 7 version 7.2.2.16 (EPI INFO by CDC, USA). Participants were counted more than once for assessing the prevalence of HR-HPV, probable/possible HR-HPV, and LR-HPV if they had types from more than one group. Single infection was defined as infection with one HPV type and multiple HPV infections were defined as the detection of two or more HPV types in the same sample. Frequencies and percentages were determined for the participant characteristics. The distribution of positive cases and genotypes was analyzed. The chi-square test was used to compare proportions. Odd ratios (OR) with 95% confidence interval were calculated to evaluate the association between HPV infection and sociodemographic and behavioral characteristics. For all data analyses and relationships between variables, a value of $p < 0.05$ was considered statistically significant.

## Ethical considerations

The LCSP Ethics Committee reviewed and approved the study (resolution N˚ 110/2019). The participants signed a free and informed written consent form before inclusion in the study. No women younger than 18 years old were included in the study. Since in Paraguay, primary screening was aimed mainly at women aged 30 years and above, to detect precursor lesions of cervical cancer; special care was taken to explain the scope of this research study to the participants. The HPV tests results were communicated to the patients by trained health personnel with an extensive health education to inform women and avoid unnecessary additional anxiety

and distress, helping to understand HPV and its association with cervical cancer, as well as the importance of screening.

## Results

The average age of study participants was 22.5±2.1 years. Of these, 70.2% had university as education level. The mean age of first sexual intercourse was 16.8±2.1 years, and 33.2% reported a previous history of STI. Additionally, 59.1% of the participants reported having fewer than two sexual partners in the last year, 81.7% had no history of smoking, and 72.6% reported alcohol consumption (Table 1). A total of 208 cervical samples were analyzed, all of which yielded a positive result for the human CFTR gene, proving the DNA integrity and good sample quality. HPV DNA was detected in 114 specimens, resulting in an overall prevalence of 54.8% (95% CI 48.0–61.6).

The results were further analyzed for cross-sectional associations between HPV infection and sociodemographic, obstetric, and behavioral characteristics. We observed significant associations between HPV infection and women who reported not living with their partner (OR 3.25; 95% CI 1.53–6.88; p = 0.01), women with more than one sexual partner within the past

**Table 1. Participant characteristics and behavioral factors associated with HPV infection in young women from Paraguay.**

| Characteristic | Total | HPV Positive | HPV Negative | OR (95% CI) | p value |
|---|---|---|---|---|---|
| | N = 208 | n = 114 | n = 94 | | |
| *Age (years)* | | | | | |
| 18–19 | 26 (12.5%) | 16 (61.5%) | 10 (38.5%) | Reference | 0.45 |
| 20–21 | 38 (18.3%) | 18 (47.4%) | 20 (52.6%) | 1.78 (0.64–4.90) | |
| 22–23 | 66 (31.7%) | 40 (60.6%) | 26 (39.4%) | 1.04 (0.41–2.64) | |
| 24–25 | 78 (37.5%) | 40 (51.3%) | 38 (48.7%) | 1.52 (0.61–3.87) | |
| *Education level* | | | | | |
| Primary/Secondary | 62 (29.8%) | 32 (51.6%) | 30 (48.4%) | 0.83 (0.46–1.51) | 0.55 |
| University | 146 (70.2%) | 82 (56.2%) | 64 (43.8%) | | |
| *Live with* | | | | | |
| Parents/Friends/Alone | 170 (81.7%) | 102 (60.0%) | 68 (40.0%) | 3.25 (1.53–6.88) | **0.001** |
| Partner | 38 (18.3%) | 12 (31.8%) | 26 (68.4%) | | |
| *Age of the sexual debut* | | | | | |
| ≤ 16 years | 89 (42.8%) | 47 (52.8%) | 42 (47.2%) | 0.87 (0.50–1.51) | 0.62 |
| > 16 years | 119 (57.2%) | 67 (56.3%) | 52 (43.7%) | | |
| *Number of sexual partners within past year* | | | | | |
| 2 or more | 85 (40.9%) | 56 (65.9%) | 29 (34.1%) | 2.16 (1.22–3.83) | **0.008** |
| 0–1 | 123 (59.1%) | 58 (47.2%) | 65 (52.9%) | | |
| *Condom use* | | | | | |
| Never/occasional | 158 (76.0%) | 94 (59.5%) | 64 (40.5%) | 2.20 (1.15–4.22) | **0.02** |
| Always | 50 (24.0%) | 20 (40.0%) | 30 (60.0%) | | |
| *History of STI* | | | | | |
| Yes | 69 (33.2) | 46 (66.7) | 23 (33.3) | 2.09 (1.14–3.81) | **0.01** |
| No | 139 (66.8) | 68 (48.9) | 71 (51.1) | | |
| *Smoking status* | | | | | |
| Yes | 38 (18.3) | 6 (68.4) | 12 (31.6) | 2.02 (0.96–4.26) | 0.06 |
| No | 170 (81.7) | 88 (51.8) | 82 (48.2) | | |
| *Alcohol consumption* | | | | | 0.48 |
| Yes | 151 (72.6) | 85 (56.3) | 66 (43.7) | 1.24 (0.68–2.29) | |
| No | 57 (27.4) | 29 (50.9) | 28 (49.1) | | |

**Table 2. Frequency of single and multiple HPV infections (n = 114).**

| Type-specific HPV | HPV positive samples | |
|---|---|---|
| | N (%) | 95% CI (%) |
| 1 | 65 (57.0) | 47.9–66.1 |
| 2 | 35 (30.1) | 22.2–39.2 |
| 3 | 8 (7.0) | 2.3–11.7 |
| 4 or more | 6 (5.3) | 1.2–9.4 |

year (OR 2.16; 95% CI 1.22–3.83; p = 0.008), inconsistent condom usage (OR 2.20; 95% CI 1.15–4.22; p = 0.02), and prior non-HPV genital tract infections (OR 2.09; 95% CI 1.14–3.81; p = 0.01). The other variables were not significantly associated with HPV infection.

Among the positive samples, multiple infections with two to seven different HPV types represented 43.0% (95% CI 33.9–52.1) (Table 2) of the total identified infections.

At least one HR-HPV genotype was identified in 42.3% (95% CI 36.6–49.0) of the women included in the study. High frequencies of HR-HPV were prevalent at all ages analyzed: 18–19 [50.0%; (95% CI 30.0–70.0)]; 20–21 [40.0%; (95% CI 23.8–56.2)], 22–23 [56.5%; (95% CI 44.1–68.8)], and 24–25 [63.6%; (95% CI 52.0–75.2)] (Fig 1).

A total of 29 different viral types were identified, present in both single and multiple infections among 208 studied women (Fig 2). The most commonly detected HPV types were HPV-58 [14.9% (95% CI 8.4–21.5)], followed by HPV-16, HPV-51, and HPV-66 at similar frequencies [12.3%; (95% CI 6.3–18.3)].

The prevalence of HPV types preventable with currently available vaccines, was 8.2% for the bivalent vaccine (16/18) (95% CI 4.5–11.9), 13.0% for the quadrivalent vaccine (6/11/16/18) (95% CI 8.4–17.6), and 38.0% for the nonavalent vaccine (6/11/16/18/31/33/45/52/58) (95% CI 31.4–44.6%). Infection prevalence of HPV types against which vaccine cross-protection has been reported, such as 31/33/45, was 10.6% (95% CI 6.4–14.8) (Table 3).

## Discussion

The WHO global strategy for the elimination of cervical cancer as a public health problem proposes a multidisciplinary approach that includes community education, social mobilization, vaccination, detection, treatment, and palliative care.

HPV vaccination in combination with timely detection and appropriate treatment of cervical cancer precursor lesions is considered to have the potential to achieve the elimination of this disease, defined by the WHO as fewer than 4 new cases per 100,000 women-years [8].

With the introduction of the HPV vaccine, changes in the transmission dynamics of the virus are expected, making necessary to monitor changes in the prevalence of HPV and the patterns of HPV-associated diseases. Population-based cohort studies are the most appropriate to assess the impact of HPV vaccination; however, the feasibility of conducting these studies is limited in low- and middle-income countries, so cross-sectional surveys in sentinel populations could be a suitable alternative [8].

Paraguay, like many other countries worldwide, has incorporated the HPV vaccine into its national immunization program, requiring a virological surveillance to measure its impact. It can provide an early measure of vaccine effectiveness by quantifying the magnitude of reduction in HPV type-specific prevalence, including possible evidence of herd and cross protection and provide timely assessment of vaccination strategy. The principle of HPV prevalence surveillance is to recruit samples in age groups at risk of HPV exposure and conduct repeat-cross sectional analyses [16].

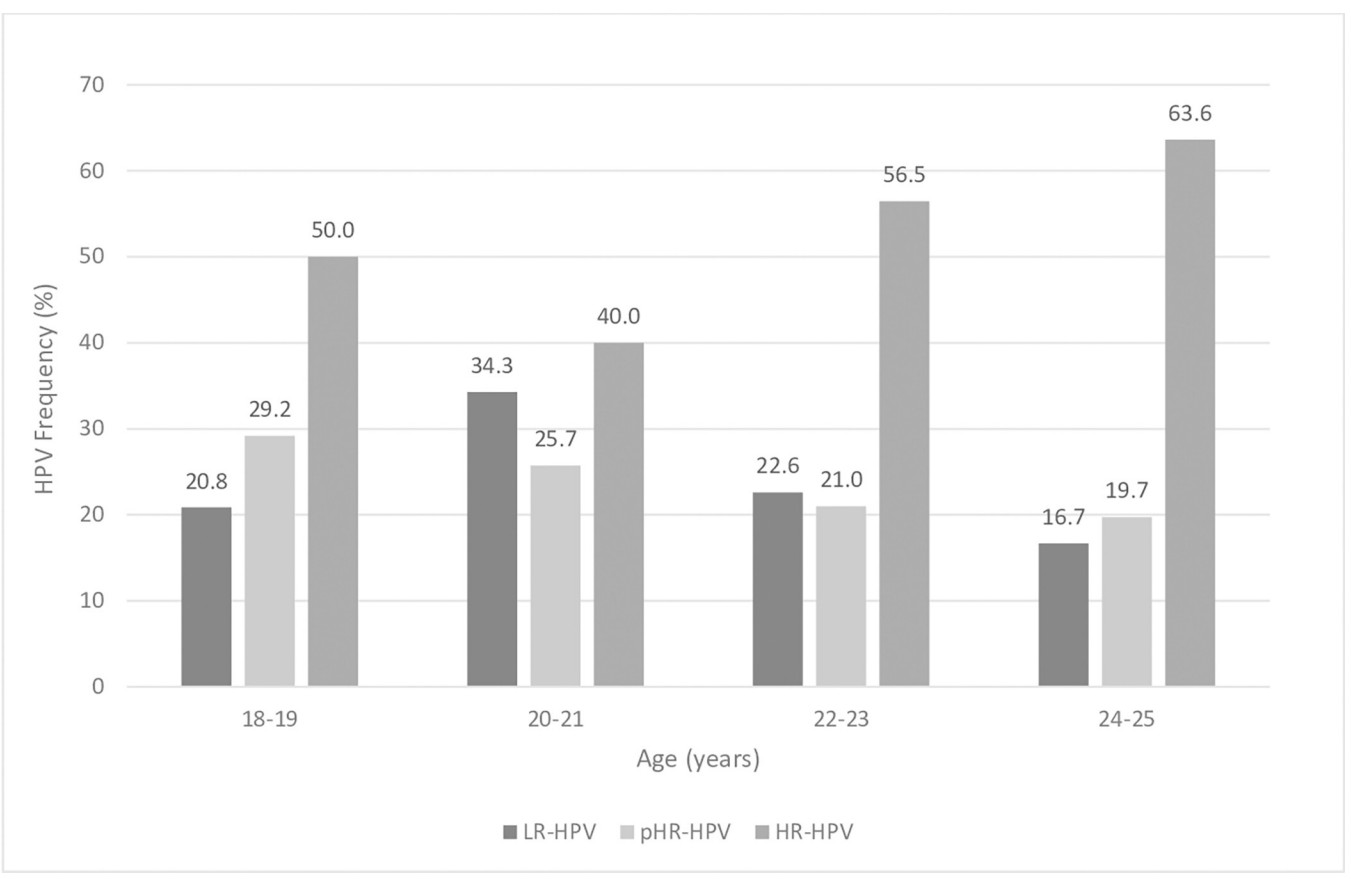

**Fig 1. Frequency of HR-HPV, pHR-HPV, and LR-HPV in young women from Paraguay (N = 208).** HR-HPV: High risk HPV types; pHR-HPV: possible high risk; and LR-HPV: low risk HPV types. HR types, including Group 1 IARC, carcinogenic to humans (HPV16, 18, 31, 33, 35, 39, 45, 51, 52, 56, 58, and 59) and Group 2A IARC, probable carcinogenic to humans (HPV68). pHR types, Group 2B IARC, possible carcinogenic to humans (HPV26, 30, 34, 53, 66, 67, 69, 70, 73, 82, 85, and 97). LR types, non-carcinogenic to humans (HPV6, 11, 40, 42, 54, 61, 71, 72, 81, 83, and 84).

In this study, we analyzed a cohort of unvaccinated sexually active women between 18–25 years of age and found an overall HPV prevalence of 54.8%, comparable to others cross-sectional studies in South American as Colombia with a prevalence of 60.3% in a same age group [17], and Argentina with a prevalence of 56.3% in 15 to 17-year-old female adolescents [18], showing the high of HPV infections in young women in the region. Our observed HPV prevalence rate is also like the 55% infection rate observed at enrollment in a longitudinal study of young women [19]. However, this rate was higher than that observed in studies in unvaccinated young women from United States (47.5% for 21–24 years) [20] and Europe, which has ranged from 25% to 33% among women aged <25 years [21–23]. Thus, the overall HPV prevalence is low in developed countries [24].

Previous studies have established sexual contact as the main route of HPV transmission. Thus, the higher HPV prevalence in women ≤25 years of age could be explained by recent first sexual contact, a condition that increases the chance of exposure to the virus [5, 24, 25].

Although the demographic data and all risk factors for acquiring HPV showed that the participants were moderate risk e.g., only 24.0% used condom, 33% reported STI, 40.9% had reported 2 or more partners, the results reported high risk HPV types. At least one type of HR-HPV was found in 42.3% of young women, with HR-HPV prevalence ranging from 18.7%–45.7% among women under 25 years old worldwide [26–33]. A similar rate (42.2%) has

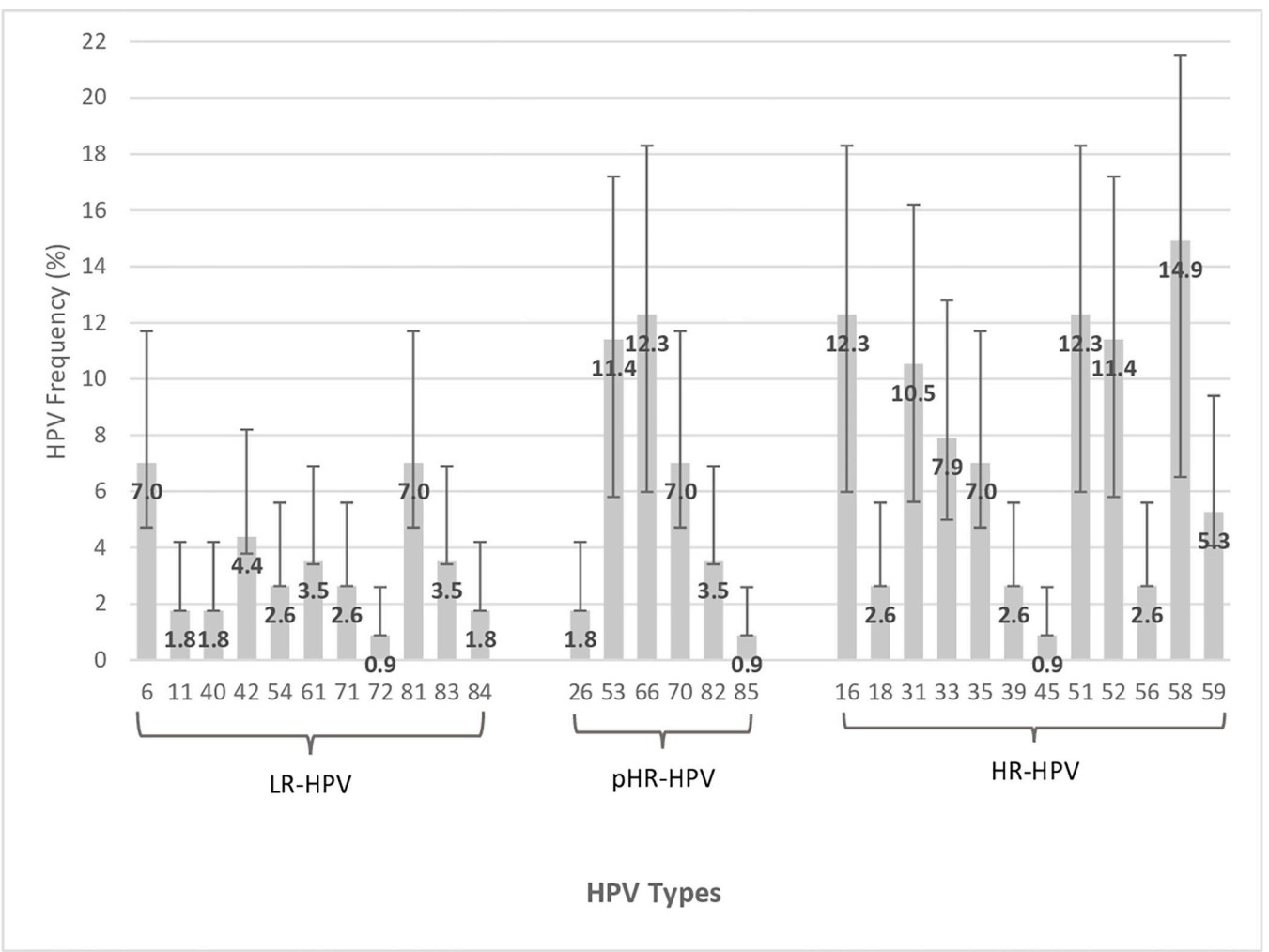

**Fig 2. Distribution of viral types among HPV positive genital samples in women with single and multiple infections (N = 208).** HR-HPV: High risk HPV types; pHR-HPV: possible high risk; and LR-HPV: low risk HPV types. HR types, including Group 1 IARC, carcinogenic to humans (HPV16, 18, 31, 33, 35, 39, 45, 51, 52, 56, 58, and 59) and Group 2A IARC, probable carcinogenic to humans (HPV68). pHR types, Group 2B IARC, possible carcinogenic to humans (HPV26, 30, 34, 53, 66, 67, 69, 70, 73, 82, 85, and 97). LR types, non-carcinogenic to humans (HPV6, 11, 40, 42, 54, 61, 71, 72, 81, 83, and 84).

**Table 3. Frequencies of vaccine-preventable HPV types in young women (N = 208).**

| HPV types | N° | % (95% CI) |
|---|---|---|
| *2-valent/4-valent/9-valent HPV vaccine types* | | |
| HPV-16 | 14 | 6.7 (3.3–10.1) |
| HPV-18 | 3 | 1.4 (0.0–3.1) |
| HPV-16/18 | 17 | 8.2 (4.5–11.9) |
| HPV-6 | 8 | 3.8 (1.2–6.5) |
| HPV-11 | 2 | 1.0 (0.0–2.3) |
| HPV-6/11 | 10 | 4.8 (1.9–7.7) |
| HPV-6/11/16/18 | 27 | 13.0 (8.4–17.6) |
| HPV-6/11/16/18/31/33/45/52/58 | 79 | 38.0 (31.4–44.6) |
| *Vaccine cross-protection types* | | |
| HPV-31/33/45 | 22 | 10.6 (6.4–14.8) |

been observed in regional surveys conducted in Colombia and Argentina [17, 18]. However, an elevated HR-HPV infection rate in young women does not necessarily lead to the development of precancerous lesions and cancer. The decreasing frequency of infection with age has been demonstrated in national, regional, and global studies, indicating the self-limited nature and control of the infection by the action of the host's immune system in most cases [24, 25, 34, 35].

As expected, sexual behavior is associated with an increased risk of HPV infection, as well as the number of sexual partners, new sexual partners, non-use of condoms, and a history of other sexual infections. A high number of sexual partners is the most important risk factor for HPV positivity, and newly acquired sexual partners further increase this risk [5, 25, 36]. Condom use may help reduce the risk of cervical and vulvovaginal HPV infection [37]. Other STI like *Chlamydia trachomatis* have been associated with cell inflammatory processes in the cervix, facilitating HPV entry and reducing its elimination [25, 38].

We identified multiple infections in 43.0% of all cases, as previously reported in younger groups of women [39, 40]. A few studies have associated multiple HPV infections with an increased risk of cervical intraepithelial neoplasia [41, 42]. However, other studies found no significant difference in the development of cervical intraepithelial neoplasia and cancer in women with single or multiple infections [43, 44]. The total risk of precancer in women infected with several HPV types was increased compared to women infected with a single HPV type [45]. It is not clear whether the risk from multiple infections is greater than the sum of the risks posed by individual HPV types. However, despite the presence of multiple viral types in a sample, only one HR-HPV leads to potential malignant transformation [46].

HPV types differ in their prevalence worldwide in diverse populations based on age range, cytologic condition, laboratory method used, and the interaction between HPV types and host immunogenetic factors. We identified 29 distinct HPV types, with HPV-58 being the most detected in 14.9% samples, followed by HPV-16, 51, 52, 31, and 33 among the HR-HPV types. The five most frequent types in women with normal cytology worldwide are HPV-16, 18, 31, 58, and 52, and in South America are HPV-16, 58, 18, 45, and 31 [24]. HPV-58, 51, and 52 are relatively abundant in Costa Rica, followed by HPV-31 and 18 [43]. The most common high-risk viral types in Chile are HPV-16, 56, 31, 58, 59, 18, and 52 [28] in Brazil are HPV-52, 16, 58, 51, 31, and 59 [47]. In Colombia, infections with HPV-16 are the most frequent, followed by HPV-58 [39]. Likewise, HPV-16, 58, and 31 have been reported as the most predominant types in Paraguayan women with normal cytology [34].

It is established that persistent infections by HPV-16 have a higher probability of causing precancerous lesions than infection with other viral types [48, 49]. Women with normal cytology and persistent HPV-16 infection have an approximately 30% chance of developing high-grade lesions in a 12-month period [50]. HPV-58 is also included in high-risk group and has been frequently reported in cervical cancer and high-grade lesions in several regions of the world [51], such as Asia [52, 53], Mexico [54, 55], Costa Rica [43], and Brazil [56]. Given the wide circulation of HPV-58 in the population studied and the evidence of its oncogenic potential, particular attention should be given to HPV-58 infection.

We found that 8.2% of the women were infected with HPV-16/18 that are included in the bivalent vaccine, close to that reported in similar populations from Colombia (12.9%) [17] and the United States (10.0%) [57]. Infections with types included in the quadrivalent vaccine (HPV-6/11/16/18) were detected in 13.0% of women, comparable to a nationally representative United States survey of females conducted from 2003 to 2006 with a prevalence of 18.5% in the 20–24-year-old group [58], but less than that reported in Argentine adolescents (22.5%) [18]. HPV 31, 33, and 45 were detected in 10.6% of women. These infections could also be

prevented, as the bivalent and quadrivalent vaccines provide cross-protection against these genotypes [10].

HPV-58, not targeted by the vaccine currently in use in Paraguay's vaccination program was the most prevalent type in this studied group. Consequently, the incidence and risk of non-vaccine HPV types must be considered to assess the impact of the quadrivalent vaccine used in the country in the national immunization program.

Have been reported women immunized with quadrivalent vaccine at young age who still developed cervical intraepithelial neoplasia with evidence that these lesions are commonly associated with HPV types not targeted by this vaccine, but some of the HPV types associated with these lesions in vaccinated women are targeted by the Gardasil-9 [59].

As we know, the nonavalent vaccine contains five additional oncogenic types (HPV-31, 33, 45, 52, and 58) and according to the current data, this vaccine could prevent 38.0% of HPV infections identified in this study. Therefore, the high prevalence of HPV types targeted by nonavalent HPV vaccines could support its introduction in the national immunization program to achieve greater coverage, considering that the preventable proportion of cervical cancers is estimated to increase to 90% [60].

The present study had some limitations as the small sample size of population studied, for this reason the results cannot be extrapolated to the total young women from Paraguay. The recruitment strategy by social networks and flyers at local health centers and higher education institutes may have induced a selection bias. Although this was not a population-based study, the results are particularly important, providing information regarded HPV prevalence in groups that are not often included in epidemiological surveys or cervical cancer prevention programs. We know that the population-based cohort studies are the most appropriate to assess the impact of HPV vaccination; however, the feasibility of conducting these studies is limited in low resources countries, thus cross-sectional surveys in sentinel populations could be a suitable alternative.

For hence this study even its important limitations, such as the small sample size and the convenience recruitment strategy, the results are particularly important, providing information regarding HPV prevalence in groups that are no often included in national cervical cancer prevention programs. These data provide the baseline against which to compare future HPV prevalence of vaccine and non-vaccine-types, allowing the short-term estimation of the impact of the national HPV immunization program.

Even though several studies provide evidence that genital HPV infection is very common in young sexually active women and in most cases is a transient and self-limited infection, the evaluation of overall HPV prevalence in this population has been proposed as a short-term endpoint for determining the impact of HPV vaccination in surveillance programs.

This research could provide key information to healthcare policy makers about the importance of increasing HPV vaccination coverage in national immunization programs. The results should be combined with other studies in cohorts of vaccinated women to investigate the reduction of types 16 and 18, as well as their possible replacement by other genotypes, thus evaluating the impact of the implementation of HPV vaccination in Paraguay and compare the results with other countries.

In addition, these data can improve the acceptance of the HPV vaccine in the Paraguayan population, highlighting the high prevalence of HPV, especially the high-risk types, observed in unvaccinated young women. Increased public awareness of the benefits of HPV vaccination for cervical cancer prevention could increase public support for immunization.

In conclusion, this is the first report on the prevalence of HPV and its genotype distribution, including HPV types targeted by current vaccines, in unvaccinated young sexually active women from Paraguay. These data provide the baseline against which to compare future HPV

prevalence of vaccine and non-vaccine-types, allowing the estimation of the impact of the national HPV immunization program. Surveillance and monitoring of changes in viral transmission that lead to modification of HPV type-specific prevalence and the patters of HPV-associated disease are crucial to progress toward the global elimination of cervical cancer.

## Author Contributions

**Conceptualization:** María Liz Bobadilla, Vanessa Salete de Paula.

**Data curation:** María Liz Bobadilla.

**Formal analysis:** María Liz Bobadilla, Gerardo Deluca.

**Investigation:** María Liz Bobadilla, Verónica Villagra, Violeta Ortiz.

**Methodology:** María Liz Bobadilla, Violeta Ortiz, Gerardo Deluca, Vanessa Salete de Paula.

**Supervision:** Verónica Villagra.

**Validation:** Verónica Villagra, Gerardo Deluca.

**Writing – original draft:** María Liz Bobadilla.

**Writing – review & editing:** María Liz Bobadilla, Vanessa Salete de Paula.

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
