## [Decision Letter · Decision Letter 0]

11 Nov 2022

PONE-D-22-28769HIGH PREVALENCE AND CO-INFECTION OF HIGH-RISK HUMAN PAPILLOMAVIRUS GENOTYPES AMONG UNVACCINATED YOUNG WOMEN FROM PARAGUAYPLOS ONE

Dear Dr. Bobadilla,

Thank you for submitting your manuscript to PLOS ONE. After careful consideration, we feel that it has merit but does not fully meet PLOS ONE’s publication criteria as it currently stands. Therefore, we invite you to submit a revised version of the manuscript that addresses the points raised during the review process.

Dear authors, 

the topic of the present article titled “HIGH PREVALENCE AND CO-INFECTION OF HIGH-RISK HUMAN PAPILLOMAVIRUS GENOTYPES AMONG UNVACCINATED YOUNG WOMEN FROM PARAGUAY” is very interesting, the paper and the aim falls within the scope of the journal but the article needs major improvements.

The introduction, material and method section and tables should be modified and improved. 

The manuscript should be organized better and English should be improved.

I suggest improving the manuscript with the reviewers' comments.

We look forward to receiving your revised manuscript.

Kind regards,

Andrea Giannini

Academic Editor

PLOS ONE

Journal Requirements:

2. Thank you for submitting the above manuscript to PLOS ONE. During our internal evaluation of the manuscript, we found significant text overlap between your submission and previous work in the Introduction section. We would like to make you aware that copying extracts from previous publications, especially outside the methods section, word-for-word is unacceptable. In addition, the reproduction of text from published reports has implications for the copyright that may apply to the publications. Please revise the manuscript to rephrase the duplicated text, cite your sources, and provide details as to how the current manuscript advances on previous work. Please note that further consideration is dependent on the submission of a manuscript that addresses these concerns about the overlap in text with published work. We will carefully review your manuscript upon resubmission and further consideration of the manuscript is dependent on the text overlap being addressed in full. Please ensure that your revision is thorough as failure to address the concerns to our satisfaction may result in your submission not being considered further.

3. Please include a complete copy of PLOS’ questionnaire on inclusivity in global research in your revised manuscript. Our policy for research in this area aims to improve transparency in the reporting of research performed outside of researchers’ own country or community. The policy applies to researchers who have travelled to a different country to conduct research, research with Indigenous populations or their lands, and research on cultural artefacts. The questionnaire can also be requested at the journal’s discretion for any other submissions, even if these conditions are not met.  Please find more information on the policy and a link to download a blank copy of the questionnaire here: https://journals.plos.org/plosone/s/best-practices-in-research-reporting. Please upload a completed version of your questionnaire as Supporting Information when you resubmit your manuscript.

4. Please provide additional details regarding participant consent. In the ethics statement in the Methods and online submission information, please ensure that you have specified what type you obtained (for instance, written or verbal, and if verbal, how it was documented and witnessed). If your study included minors, state whether you obtained consent from parents or guardians. If the need for consent was waived by the ethics committee, please include this information.

6. Thank you for stating the following financial disclosure: 

"Funding information: This  work  was  supported  by the National Council of Science and Technology (CONACYT) through the PROCIENCIA Program with resources from the Fund for Excellence in Education and Research (FEEI) (RESERCH 15-INV-200) and  by the Research, Education and Biotechnologies Applied to Health (IEBAS) Project – MERCOSUR  Structural  Convergence  Fund (FOCEM), COF 03-11."

7. We note that you have indicated that data from this study are available upon request. PLOS only allows data to be available upon request if there are legal or ethical restrictions on sharing data publicly. For more information on unacceptable data access restrictions, please see http://journals.plos.org/plosone/s/data-availability#loc-unacceptable-data-access-restrictions. 

Reviewers' comments:

Reviewer's Responses to Questions

**Comments to the Author**

1. Is the manuscript technically sound, and do the data support the conclusions?

Reviewer #1: Partly

Reviewer #2: Partly

Reviewer #3: Partly

Reviewer #4: Partly

Reviewer #5: Yes

2. Has the statistical analysis been performed appropriately and rigorously? 

Reviewer #1: Yes

Reviewer #2: Yes

Reviewer #3: Yes

Reviewer #4: Yes

Reviewer #5: Yes

3. Have the authors made all data underlying the findings in their manuscript fully available?

Reviewer #1: Yes

Reviewer #2: No

Reviewer #3: Yes

Reviewer #4: No

Reviewer #5: Yes

4. Is the manuscript presented in an intelligible fashion and written in standard English?

Reviewer #1: No

Reviewer #2: Yes

Reviewer #3: Yes

Reviewer #4: Yes

Reviewer #5: Yes

5. Review Comments to the Author

Reviewer #1: Thank you for including me in reviewing this manuscript. The article is that “HIGH PREVALENCE AND CO-INFECTION OF HIGH-RISK HUMAN PAPILLOMAVIRUS GENOTYPES AMONG UNVACCINATED YOUNG WOMEN FROM PARAGUAY”. The sample size is small. The current study does not deliver anything significant new or outbreaking in this matter; at this point, all of this, justifies rejection of the manuscript.

1. The introduction is long. Authors should abbreviate some places by combining them. Otherwise, the interest in the subject and the flow of the subject are scattered. For example, sections on vaccines and HPV can be shortened. Places that indicate the purpose of the subject can be highlighted.

2. There are minimal grammer errors. I recommend that a native English speaker review the article or use a professional language editing service.

3. ”National primary screening is mainly aimed

at women aged 30 years and above; younger women does not frequently attend health services

for screening tests.” What is this program like, how is it done?

4. You mentioned the vaccination program that started in 2013. What is this program?

5. “which was similar to that reported in other…” Is the age group of all the other studies mentioned in this paragraph the same?

6. As you know, HPV can be eliminated after a certain period of time. There may be elimination, especially under the age of 25. What do the authors think about this and about screening in the early age group?

Reviewer #2: The authors conducted an HPV prevalence and genotyping study among unvaccinated females using screening samples in women aged 18-25 years. The data are important because they establish a baseline for monitoring impact of Paraguay’s HPV vaccination program. The study appears to be internally valid and most of the analyses seem to be appropriate. Some edits could improve the readability and further focus the paper on the important information.

Although the manuscript is generally well-written, additional attention to English grammar is warranted.

Abstract:

First sentence stating “several young women remain unvaccinated” is quite confusing. Isn’t it true that at the time of this study, women in the age group included were never vaccine eligible? Quadrivalent vaccine should be moved from second sentence to first (if virologic surveillance is important, then it would be true for a vaccination program with any formulation). Methods in abstract are almost non-existent – please include some information on the recruitment procedures, including how epidemiologic data and specimen were collected and some basics of the typing assay, such as number of types detected.

Introduction:

The introduction is quite long and could be trimmed considerably to focus more on essential information needed to understand the context of the paper. Some examples are listed below.

The first two sentences of the paper suggest the main reason young women are susceptible to STIs is immaturity of genital tract. While this may be a contributor, the initiation of sexual activity and non-monogamous relationships should not be ignored. Those first two sentences could be deleted – “HPV is the most common STI” is a better start. The second paragraph is largely extraneous and most if not all of it could be deleted. Line 53 introduces the term “high-risk types,” which needs to be defined, as it was not introduced in the previous paragraph about the categories of HPV types.

The introduction really needs a more complete description of the vaccination program. There is a long paragraph on page 4 beginning with line 64 describing all the internationally approved vaccines, and easily lost at the end of this paragraph is one sentence about Paraguay: “In Paraguay, the quadrivalent Gardasil vaccine was introduced to 10-year-old girls in 2013.” The international information could be shortened, and more information on the Paraguay program added. How many doses of Gardasil do girls receive? In what setting (e.g., schools or primary care)? What has the coverage been? Doing the math, with the study conducted in 2020-21, the females included in the prevalence study had mainly not been vaccine eligible, is this correct? This needs to be spelled out for the reader. On first read, it seemed like the authors had screened out HPV-vaccinated women, which raised questions about vaccination coverage and known herd effects from HPV vaccination. This may still be an issue in the youngest women, and should be addressed in the discussion. It is not a real problem with internal validity, but it needs to be clear what exactly was done and why.

Terminology with different meanings has been used interchangeably (e.g., on page 5 “impact” in line 90, “efficacy” in line 93, and “effectiveness” in line 95). Impact is the correct term for what this study provides a baseline to assess. Similarly, throughout the paper, the authors refer frequently to vaccines “containing” different HPV types; as there is no virus in the vaccines, it may be more appropriate to say the vaccines “target” different HPV types.

Materials and methods:

Study Population:

Please provide more clarity regarding recruitment methods. Do women in this age group normally attend the LCSP for reasons other than study participation? Were women recruited from the LCSP or from other community sites? Was any incentive provided? Were women given results of the HPV testing, and if so, how were they counseled? Was the study actually conducted on “randomly selected samples”, as stated on page 5 in line 103, or were a convenience sample of women recruited to participate in this study? These details are important for replicating study methods in a future evaluation of impact.

Statistical analysis:

In the statistical analysis section on page 7 in lines 150-156, please describe the HPV type groupings, and how multiple types were accounted for. For example, if a woman had HPV 18 and HPV 58 detected, would she be included in the numerator of both types? In Figure 1, if a woman had both a HR and a LR type detected, where would she be represented?

Figure 1 is potentially misleading because it shows proportion positive for the different type groups out of 100% -- this makes it appear that the prevalence of HR-HPV is higher in older age groups, but that may not be true, and the prevalence is not shown. Consider remaking this figure so that the height of the bars represents total prevalence in each age group.

On page 7 in line 151, consider not using “qualitative variables” in the description of the methods. “Participant characteristics” would be a better descriptor.

Some analyses presented in the results (e.g. odds ratios and 95% CIs) are not described in the stastistical analysis section.

Results:

Table 1 and the corresponding text description are inconsistent. For example, the text refers to percentage who were students without employment, but the table does not include employment status. The text refers to percentage with at least high school education, but the table combines primary and secondary. Table 1 does not show results by age; age is usually one of the strongest predictors of HPV prevalence, and this basic descriptive information needs to be provided.

Consider not reporting odds ratios since this may not be the most appropriate measure of association for this study. Since HPV is not a rare outcome, odds ratios overestimate the strength of the association. Consider not reporting a measure of association at all (relying on comparing percentages with p-values) or consider using prevalence ratios instead. Regardless, the description of this measure of association should be included in the statistical analyses section.

In table 3, please specify N. HPV6 prevalence appears to be off by .1.

In the description of figure 2 on page 11 beginning with line 200, please be clear about how prevalence is determined for single and multiple infections. What is the denominator? Is it 208?

In the legend of figure 2 on page 11 in line 210, HPV 72 LR is not mentioned.

Discussion:

Generally, the discussion could be trimmed to focus on the most salient information. For example, the paragraph on multiple types that starts on page 13 with line 245 may be extraneous. The last paragraph before the conclusion beginning on page 15 with line 295 is confusing, with discussion expanding to other STIs. This paragraph may not be needed as well. A very high number of references is cited (76), and the authors should focus on those that are most important to the topic of establishing a baseline HPV prevalence in Paraguay.

On page 14 beginning with line 271, consider adding more context to the discussion of HPV 58. While HPV 58 has a high prevalence in Paraguay and in other countries worldwide, before suggesting it be given equal consideration to HPV-16, its degree of carcinogenicity should be considered. Most HPV-attributable cancers are caused by HPV-16 and 18. International studies found HPV-58 in 2.2% (de Sanjose JNCI Cancer Spectr 2018) and 4.4% (Guan et al., IJC 2012) of invasive cervical cancers, far below the contributions of the high-risk types targeted by the quadrivalent vaccine.

On page 15 in line 285, it is confusing to state that HPV 58 is not vaccine preventable (it is targeted by Gardasil-9); better to say it is not targeted by the vaccine currently in use in Paraguay’s vaccination program.

The discussion does not appear to include a paragraph on limitations of the study. It is important to at least acknowledge limitations of the study population not being representative of the population of Paraguay or Asuncion and how recruitment of a comparable population for a follow-up study to measure vaccine impact would be accomplished.

On page 16, in the conclusion in lines 308-309, consider deleting “Considering the high prevalence, multiple infections, and non-vaccine-type HPV infections“ from the second sentence. The data provide a baseline regardless of these considerations.

Reviewer #3: 1) The title the authors picked is an interesting and pioneer base line focus of interest specific to Paraguay. Authors used simple random sampling to select 208 women aged 18-25. Do you have an explanation or a local reference for the statement on line 106 "younger women..."? Secondly, what is the conclusion you have drawn from this research? Furthermore, what are your recommendations?

2) Although looking grossly sound, I have one questions regarding the result/analysis section. Why is the summation of those with History of STI and Negative test not 94; it is 91 in your case?

Overall, it is a very well written manuscript discussing a hot issue mainly concerning developing countries like Paraguay. It is a good initiative but the conclusion and recommendation needed further elaboration than short statements.

Reviewer #4: This study aimed to estimate the type-specific HPV frequency in unvaccinated women aged 18–25 years in the metropolitan area of Asuncion as a baseline for monitoring the HPV vaccination program

I have reviewed the manuscript and would like to suggest the following changes.

Discussion :

The first paragraph should mention the noteworthy findings of the study. The authors should compare and contrast these findings with other studies in subsequent paragraphs.

Also, author should add a paragraph regarding limitations and future implication of this study.

the conclusion should be a separate paragraph

the references are very numerous, it may be necessary to reduce a little

Reviewer #5: The paper was on the HIGH PREVALENCE AND CO-INFECTION OF HIGH-RISK HUMAN PAPILLOMAVIRUS GENOTYPES AMONG UNVACCINATED YOUNG WOMEN FROM PARAGUAY. The results showed important information for public health.

However, as the title was on prevalence, there were no mentioned on the estimation of sample size as N=208 was small. the demographic data showed that the participants were actually moderate risk eg. only 24.0 % used condom, 33% reported STI, 40.9% had reported 2 or more partners. These should add in the discussion, all these are risk factors for acquiring HPV. That's why the results reported quite high high risk HPV types.

6. PLOS authors have the option to publish the peer review history of their article (what does this mean?). If published, this will include your full peer review and any attached files.

Reviewer #1: **Yes: **Dr. Burak Bayraktar

Reviewer #2: No

Reviewer #3: **Yes: **Dr. Kaleab A. Betru

Reviewer #4: No

Reviewer #5: No

---

## [Author Response · Author response to Decision Letter 0]

23 Dec 2022

Taking into account the important suggestions proposed by the reviewers and seeking to respond to each of them, all the points were answered in the letter to the reviewers, please refer to this attached file RESPONSE TO REVIEWERS.

---

## [Decision Letter · Decision Letter 1]

25 Jan 2023

PONE-D-22-28769R1HIGH PREVALENCE AND CO-INFECTION OF HIGH-RISK HUMAN PAPILLOMAVIRUS GENOTYPES AMONG UNVACCINATED YOUNG WOMEN FROM PARAGUAYPLOS ONE

Dear Dr. Bobadilla,

Thank you for submitting your manuscript to PLOS ONE. After careful consideration, we feel that it has merit but does not fully meet PLOS ONE’s publication criteria as it currently stands. Therefore, we invite you to submit a revised version of the manuscript that addresses the points raised during the review process.

We look forward to receiving your revised manuscript.

Kind regards,

Andrea Giannini

Academic Editor

PLOS ONE

Journal Requirements:

Additional Editor Comments:

Dear authors,

the manuscript it has now been evaluated by our experts and they have recommended that minor changes be made to the submission.

Please improving the manuscript with the reviewers' comments.

Reviewers' comments:

Reviewer's Responses to Questions

**Comments to the Author**

1. If the authors have adequately addressed your comments raised in a previous round of review and you feel that this manuscript is now acceptable for publication, you may indicate that here to bypass the “Comments to the Author” section, enter your conflict of interest statement in the “Confidential to Editor” section, and submit your "Accept" recommendation.

Reviewer #1: (No Response)

Reviewer #4: All comments have been addressed

Reviewer #6: (No Response)

Reviewer #7: (No Response)

2. Is the manuscript technically sound, and do the data support the conclusions?

Reviewer #1: Partly

Reviewer #4: Partly

Reviewer #6: Partly

Reviewer #7: Yes

3. Has the statistical analysis been performed appropriately and rigorously? 

Reviewer #1: Yes

Reviewer #4: No

Reviewer #6: Yes

Reviewer #7: Yes

4. Have the authors made all data underlying the findings in their manuscript fully available?

Reviewer #1: Yes

Reviewer #4: Yes

Reviewer #6: Yes

Reviewer #7: Yes

5. Is the manuscript presented in an intelligible fashion and written in standard English?

Reviewer #1: No

Reviewer #4: Yes

Reviewer #6: Yes

Reviewer #7: Yes

6. Review Comments to the Author

Reviewer #1: The authors made several corrections. But as I said in my first review:The current study does not deliver anything significant new or outbreaking in this matter.

Reviewer #4: (No Response)

Reviewer #6: I read with great interest the manuscript, which falls within the aim of this Journal. In my honest opinion, the topic is interesting enough to attract the readers’ attention. Nevertheless, the authors should clarify some points and improve the discussion, as suggested below.

Authors should consider the following recommendations:

- Manuscript should be further revised in order to correct some typos and improve style.

- To date, several lines of evidence support the possibility to use specific biomarkers to identify early-stage cervical cancer and, in this way, offer a better prognosis to the patients. This point deserves to be discussed, referring to: PMID: 28918603; PMID: 35549629.

- I would recommend to stress novel pieces of evidence about high-risk HPV-negative high-grade cervical dysplasia, which seems to have more favorable outcomes than patients with documented high-risk-HPV infection (PMID: 35742340; PMID: 33514481).

Reviewer #7: I read with great interest the Manuscript titled "HIGH PREVALENCE AND CO-INFECTION OF HIGH-RISK HUMAN PAPILLOMAVIRUS GENOTYPES AMONG UNVACCINATED YOUNG WOMEN FROM PARAGUAY " which falls within the aim of the Journal.

In my honest opinion, the topic is interesting enough to attract the readers’ attention. Methodology is accurate and conclusions are supported by the data analysis. Nevertheless, authors should clarify some point and improve the discussion citing relevant and novel key articles about the topic.

-The introduction should be extended and completed. I find interesting a reference to the efforts made for the prevention and early diagnosis of gynecological cancers (see PMID: 36141217).

- Although it is a retrospective analysis, inclusion/exclusion criteria should be better clarified by extending their description.

- Discussions can be expanded and improved by citing relevant articles (I suggest authors to read and insert in references the following article PMID: 35742340).

Considered all this points, I think it could be of interest for the readers and, in my opinion, it deserves the priority to be published after minor revisions.

7. PLOS authors have the option to publish the peer review history of their article (what does this mean?). If published, this will include your full peer review and any attached files.

Reviewer #1: **Yes: **Burak Bayraktar

Reviewer #4: No

Reviewer #6: No

Reviewer #7: **Yes: **Tullio Golia D'Augè

---

## [Author Response · Author response to Decision Letter 1]

1 Mar 2023

REVIEW COMMENTS TO THE AUTHOR

1) REVIEWER #1: The authors made several corrections. But as I said in my first review: The current study does not deliver anything significant new or outbreaking in this matter.

 We respect your opinion, but as you know the incidence of cervical cancer in Paraguay is higher than other countries in the region and an important component of its prevention is the HPV vaccination. So, Paraguay, like many other countries worldwide, has incorporated the HPV vaccine into its national immunization program, requiring a virological surveillance to monitor changes in the virus transmission. We know that the population-based cohort studies are the most appropriate to assess the impact of HPV vaccination; however, the feasibility of conducting these studies is limited in low resources countries, thus cross-sectional surveys in sentinel populations could be a suitable alternative.

 For hence this study even its important limitations, such as the small sample size and the convenience recruitment strategy, the results are particularly important, providing information regarding HPV prevalence in groups that are no often included in national cervical cancer prevention programs. These data provide the baseline against which to compare future HPV prevalence of vaccine and non-vaccine-types, allowing the short-term estimation of the impact of the national HPV immunization program. 

 These explanations were added in the discussion. 

 Lines 342-351. We know that the population-based cohort studies are the most appropriate to assess the impact of HPV vaccination; however, the feasibility of conducting these studies is limited in low resources countries, thus cross-sectional surveys in sentinel populations could be a suitable alternative.

For hence this study even its important limitations, such as the small sample size and the convenience recruitment strategy, the results are particularly important, providing information regarding HPV prevalence in groups that are no often included in national cervical cancer prevention programs. These data provide the baseline against which to compare future HPV prevalence of vaccine and non-vaccine-types, allowing the short-term estimation of the impact of the national HPV immunization program.

2) REVIEWER #4: (No Response)

3) REVIEWER #6: I read with great interest the manuscript, which falls within the aim of this Journal. In my honest opinion, the topic is interesting enough to attract the readers’ attention. Nevertheless, the authors should clarify some points and improve the discussion, as suggested below. Authors should consider the following recommendations:

 - Manuscript should be further revised in order to correct some typos and improve style.

 Thank you for your observation. The text was revised.

 - To date, several lines of evidence support the possibility to use specific biomarkers to identify early-stage cervical cancer and, in this way, offer a better prognosis to the patients. This point deserves to be discussed, referring to: PMID: 28918603; PMID: 35549629. 

 - I would recommend to stress novel pieces of evidence about high-risk HPV-negative high-grade cervical dysplasia, which seems to have more favorable outcomes than patients with documented high-risk-HPV infection (PMID: 35742340; PMID: 33514481).

 We appreciate your suggestions, we read the mentioned references and we added some important points in the introduction:

 Lines 63 – 69. Cervical cancer is a disease which reflects inequities among different populations depending on the availability of a national vaccination program and population-based cervical cancer screening, and access to quality treatment. In 2020, the World Health Organization (WHO) called for action to eliminate cervical cancer as a public health problem. 

With three key strategies and clear 2030 targets: 90% of girls fully vaccinated with HPV vaccine by 15 years of age, 70% of women screened using a high-performance test by 35 years of age and again by 45 years, and 90% of women identified with cervical disease treated [8]. 

 Line 70. The primary prevention approach focuses on preventing disease before it develops.

 Lines 79 – 92. Secondary prevention, based on cervical screening program, is widely implemented to prevent. The adoption of Pap smear and HPV testing resulted in a significant increase in the diagnosis of cervical dysplasia and a significant decrease in cervical cancer, mainly in more developed countries. New screening strategies, such as HPV self-sampling, and the use of artificial intelligence in colposcopy assessment, must be disseminated in a near future [11].

Some markers, including p16 ink4a, p16, E-cadherin, Ki67, pRb, and p53, have been found to be useful in identifying intra-epithelial lesions that are more likely to develop into invasive lesions. Others, such as CEA, SCC-Ag, CD44, were developed to detect invasive forms. These biomarkers represent a promising diagnostic, prognostic and dynamic tool and can also be easily performed in the management of cervical cancer [12]. 

And tertiary prevention in cervical cancer includes its treatment with surgery (radical hysterectomy), radiotherapy, and systemic treatments (chemotherapy, bevacizumab, and immunotherapy) [11]. Minimally invasive surgery has been studied and associated with advantages in the treatment of various gynecological cancers [13].

4)REVIEWER #7: I read with great interest the Manuscript titled "HIGH PREVALENCE AND CO-INFECTION OF HIGH-RISK HUMAN PAPILLOMAVIRUS GENOTYPES AMONG UNVACCINATED YOUNG WOMEN FROM PARAGUAY " which falls within the aim of the Journal. In my honest opinion, the topic is interesting enough to attract the readers’ attention. Methodology is accurate and conclusions are supported by the data analysis. Nevertheless, authors should clarify some point and improve the discussion citing relevant and novel key articles about the topic.

 -The introduction should be extended and completed. I find interesting a reference to the efforts made for the prevention and early diagnosis of gynecological cancers (see PMID: 36141217).

 Based on the suggested reference, we have added information on new screening strategies and certain biological markers that may be useful in early diagnosis and prognosis of cervical dysplasia, and new approaches to the treatment pre-cancerous lesions and cervical cancer. 

 Lines 79 – 92. Secondary prevention, based on cervical screening program, is widely implemented. The adoption of Pap smear and HPV testing resulted in a significant increase in the diagnosis of cervical dysplasia and a significant decrease in cervical cancer, mainly in more developed countries. New screening strategies, such as HPV self-sampling, and the use of artificial intelligence in colposcopy assessment, must be disseminated in a near future [11].

Some markers, including p16 ink4a, p16, E-cadherin, Ki67, pRb, and p53, have been found to be useful in identifying intra-epithelial lesions that are more likely to develop into invasive lesions. Others, such as CEA, SCC-Ag, CD44, were developed to detect invasive forms. These biomarkers represent a promising diagnostic, prognostic and dynamic tool and can also be easily performed in the management of cervical cancer [12]. 

And tertiary prevention in cervical cancer includes its treatment with surgery (radical hysterectomy), radiotherapy, and systemic treatments (chemotherapy, bevacizumab, and immunotherapy) [11]. Minimally invasive surgery has been studied and associated with advantages in the treatment of various gynecological cancers [13].

 - Although it is a retrospective analysis, inclusion/exclusion criteria should be better clarified by extending their description.

 We explained more clearly the inclusion/exclusion criteria.

 Lines 114 – 117. Sexually active women aged 18–25 years, who had not been vaccinated against HPV were invited for screening through social networks and flyers at local health centers and higher education institutes. Exclusion criteria were pregnancy, nonsexual onset at the time of the study and have been received HPV vaccine.

 - Discussions can be expanded and improved by citing relevant articles (I suggest authors to read and insert in references the following article PMID: 35742340).

Considered all this points, I think it could be of interest for the readers and, in my opinion, it deserves the priority to be published after minor revisions.

 Discussions have been expanded and improved as suggested, with the follow changes: 

 Lines 235 – 253. The WHO global strategy for the elimination of cervical cancer as a public health problem proposes a multidisciplinary approach that includes community education, social mobilization, vaccination, detection, treatment, and palliative care. 

HPV vaccination in combination with timely detection and appropriate treatment of cervical cancer precursor lesions is considered to have the potential to achieve the elimination of this disease, defined by the WHO as fewer than 4 new cases per 100,000 women-years [8].

With the introduction of the HPV vaccine, changes in the transmission dynamics of the virus are expected, making necessary to monitor changes in the prevalence of HPV and the patterns of HPV-associated diseases. Population-based cohort studies are the most appropriate to assess the impact of HPV vaccination; however, the feasibility of conducting these studies is limited in low- and middle-income countries, so cross-sectional surveys in sentinel populations could be a suitable alternative [8].

Paraguay, like many other countries worldwide, has incorporated the HPV vaccine into its national immunization program, requiring a virological surveillance to measure its impact. It can provide an early measure of vaccine effectiveness by quantifying the magnitude of reduction in HPV type-specific prevalence, including possible evidence of herd and cross protection and provide timely assessment of vaccination strategy. The principle of HPV prevalence surveillance is to recruit samples in age groups at risk of HPV exposure and conduct repeat-cross sectional analyses [16].

 Lines 326 – 329. Have been reported women immunized with quadrivalent vaccine at young age who still developed cervical intraepithelial neoplasia with evidence that these lesions are commonly associated with HPV types not targeted by this vaccine, but some of the HPV types associated with these lesions in vaccinated women are targeted by the Gardasil-9 [59].

 Lines 332 – 335. Therefore, the high prevalence of HPV types targeted by nonavalent HPV vaccines could support its introduction in the national immunization program to achieve greater coverage, considering that the preventable proportion of cervical cancers is estimated to increase to 90% [60].

 Lines 342 – 351. We know that the population-based cohort studies are the most appropriate to assess the impact of HPV vaccination; however, the feasibility of conducting these studies is limited in low resources countries, thus cross-sectional surveys in sentinel populations could be a suitable alternative.

For hence this study even its important limitations, such as the small sample size and the convenience recruitment strategy, the results are particularly important, providing information regarding HPV prevalence in groups that are no often included in national cervical cancer prevention programs. These data provide the baseline against which to compare future HPV prevalence of vaccine and non-vaccine-types, allowing the short-term estimation of the impact of the national HPV immunization program.

 Lines 356 – 365. This research could provide key information to healthcare policy makers about the importance of increasing HPV vaccination coverage in national immunization programs. The results should be combined with other studies in cohorts of vaccinated women to investigate the reduction of types 16 and 18, as well as their possible replacement by other genotypes, thus evaluating the impact of the implementation of HPV vaccination in Paraguay and compare the results with other countries.

In addition, these data can improve the acceptance of the HPV vaccine in the Paraguayan population, highlighting the high prevalence of HPV, especially the high-risk types, observed in unvaccinated young women. Increased public awareness of the benefits of HPV vaccination for cervical cancer prevention could increase public support for immunization.

 Lines 370 – 373. Surveillance and monitoring of changes in viral transmission that lead to modification of HPV type-specific prevalence and the patters of HPV-associated disease are crucial to progress toward the global elimination of cervical cancer.

---

## [Decision Letter · Decision Letter 2]

12 Mar 2023

"High prevalence and co-infection of high-risk Human Papillomavirus genotypes among unvaccinated young women from Paraguay”

PONE-D-22-28769R2

Dear Dr. Bobadilla,

We’re pleased to inform you that your manuscript has been judged scientifically suitable for publication and will be formally accepted for publication once it meets all outstanding technical requirements.

Kind regards,

Andrea Giannini

Academic Editor

PLOS ONE

Additional Editor Comments (optional):

The manuscript has been modified with the comments of the reviewers. It is now ready to be published.

Reviewers' comments:

Reviewer's Responses to Questions

**Comments to the Author**

1. If the authors have adequately addressed your comments raised in a previous round of review and you feel that this manuscript is now acceptable for publication, you may indicate that here to bypass the “Comments to the Author” section, enter your conflict of interest statement in the “Confidential to Editor” section, and submit your "Accept" recommendation.

Reviewer #6: All comments have been addressed

Reviewer #7: All comments have been addressed

2. Is the manuscript technically sound, and do the data support the conclusions?

Reviewer #6: Yes

Reviewer #7: Yes

3. Has the statistical analysis been performed appropriately and rigorously? 

Reviewer #6: Yes

Reviewer #7: Yes

4. Have the authors made all data underlying the findings in their manuscript fully available?

Reviewer #6: Yes

Reviewer #7: Yes

5. Is the manuscript presented in an intelligible fashion and written in standard English?

Reviewer #6: Yes

Reviewer #7: Yes

6. Review Comments to the Author

Reviewer #6: I carefully evaluated the revised version of this manuscript.

The authors have performed the required changes, improving significantly the quality of the paper.

Reviewer #7: Dear authors, thank you for sending me the corrected manuscript.

I read your work with great interest and pleasure. The work with the changes made after my advices and those of the other reviewers is complete and, in my opinion, ready for publication.

7. PLOS authors have the option to publish the peer review history of their article (what does this mean?). If published, this will include your full peer review and any attached files.

Reviewer #6: No

Reviewer #7: **Yes: **Tullio Golia D'Augè

---

## [Editor Report · Acceptance letter]

29 Mar 2023

PONE-D-22-28769R2 

High prevalence and co-infection of high-risk *Human Papillomavirus genotypes* among unvaccinated young women from Paraguay 

Dear Dr. BOBADILLA:

I'm pleased to inform you that your manuscript has been deemed suitable for publication in PLOS ONE. Congratulations! Your manuscript is now with our production department. 

Kind regards, 

on behalf of

Dr. Andrea Giannini 

Academic Editor

PLOS ONE